# Sex-Dependent Impairment of Endothelium-Dependent Relaxation in Aorta of Mice with Overexpression of Hyaluronan in Tunica Media

**DOI:** 10.3390/ijms24098436

**Published:** 2023-05-08

**Authors:** Karen Axelgaard Lorentzen, Raquel Hernanz, Estéfano Pinilla, Jens Randel Nyengaard, Lise Wogensen, Ulf Simonsen

**Affiliations:** 1Research Laboratory for Biochemical Pathology, Department of Clinical Medicine, Aarhus University Hospital, Aarhus University, 8000 Aarhus, Denmark; 2Department of Biomedicine, Pulmonary and Cardiovascular Pharmacology, Aarhus University, 8000 Aarhus, Denmark; 3Departamento de Ciencias Básicas de la Salud, Universidad Rey Juan Carlos, 28933 Alcorcón, Spain; 4Core Center for Molecular Morphology, Section for Stereology and Microscopy, Center for Stochastic Geometry and Advanced Bioimaging, The Department of Clinical Medicine—Stereology, Aarhus University, 8000 Aarhus, Denmark

**Keywords:** diabetes, hyaluronan, endothelial dysfunction, arterial stiffness

## Abstract

Diabetic macroangiopathy is characterized by increased extracellular matrix deposition, including excessive hyaluronan accumulation, vessel thickening and stiffness, and endothelial dysfunction in large arteries. We hypothesized that the overexpression of hyaluronan in the tunica media also led to endothelial cell (EC) dysfunction. To address this hypothesis, we investigated the following in the aortas of mice with excessive hyaluronan accumulation in the tunica media (HAS-2) and wild-type mice: EC dysfunction via myograph studies, nitric oxide (NO) bioavailability via diaminofluorescence, superoxide formation via dihydroethidium fluorescence, and the distances between ECs via stereological methods. EC dysfunction, characterized by blunted relaxations in response to acetylcholine and decreased NO bioavailability, was found in the aortas of male HAS-2 mice, while it was unaltered in the aortas of female HAS-2 mice. Superoxide levels increased and extracellular superoxide dismutase (ecSOD) expression decreased in the aortas of male and female HAS-2 mice. The EC–EC distances and LDL receptor expression were markedly increased in the HAS-2 aortas of male mice. Our findings suggest hyaluronan increases oxidative stress in the vascular wall and that together with increased EC distance, it is associated with a sex-specific decrease in NO levels and endothelial dysfunction in the aorta of male HAS-2 transgenic mice.

## 1. Introduction

Diabetic macroangiopathy is characterized by increased extracellular matrix (ECM) deposition and a changed composition around the smooth muscle cells [1], vessel thickening and stiffness, and endothelial dysfunction in large arteries [2,3]. These vascular changes accelerate atherosclerosis, and diabetes is an independent risk factor [4].

In the large arteries of diabetic animals and humans, the changed composition and increased ECM deposition involve proteoglycans and glycosaminoglycans, which are negatively charged macromolecules with a high binding affinity for positively charged apolipoproteins [5]. Hyaluronan (hyaluronic acid) is an anionic non-sulfated glycosaminoglycan that forms on the plasma membrane via a process regulated by hyaluronan synthases (HAS 1-3). Hyaluronan is markedly increased in the smooth muscle layer of the vascular wall in people with diabetes [6]. Upon binding to CD44 receptors, an increase in hyaluronan may lead to the migration and proliferation of smooth muscle cells [7]. Hyaluronan accumulation in the vascular wall may also lead to the retention of more lipids, thereby accelerating atherosclerosis development [8,9]. Indeed, in a transgenic mouse model with vascular smooth muscle overexpression of HAS-2 followed by hyaluronan accumulation, accelerated atherosclerosis occurred when this mouse was cross-bred with ApoE^−/−^ mice [10]. The HAS-2 animal model has several characteristics of early diabetic angiopathy without high plasma glucose. HAS-2 transgenic mice undergo hyaluronan accumulation in the tunica media, an increase in the strength, stiffness, an cross-sectional area of the aorta, remodeling of the ECM in the media, and a shift in smooth muscle phenotype and proliferation [10,11]. Therefore, these animals represent an excellent model for investigating endothelial dysfunction in diabetes-related aortic stiffness without considering high plasma glucose, hyperlipidemia, or other effects caused by diabetic metabolism. 

Dysfunction of the endothelial cell (EC) layer is characterized by impaired vasodilation and is an independent risk factor for cardiovascular events [12,13,14]. Endothelial dysfunction is often characterized by a shift in the nitric oxide (NO)/superoxide anion (O_2_-) balance, and is associated with a decreased response to the endothelium-dependent vasodilator, acetylcholine (ACh). Endothelial NO synthase (eNOS) is activated by the phosphorylation of eNOS-Ser^1177^, and produces NO under non-stressed conditions as a homodimer in the presence of its substrates oxygen and L-arginine, with tetrahydrobiopterin as a cofactor [15,16]. During oxidative stress, tetrahydrobiopterin is oxidized into dihydrobiopterin, which results in eNOS uncoupling [3,17]. The eNOS monomers produce O_2_- instead of NO, thus further contributing to oxidative stress [17]. Extracellular superoxide dismutase (ecSOD) is a substantial contributor to the vasculature’s resistance to oxidative stress [18], and while its absence is not troubling under normal circumstances, any form of oxidative stress induction reveals that it has an essential role in vascular maintenance [18,19,20,21,22].

In previous studies, we found increased stiffness in the aortas of HAS-2 mice [10,11], a well-known risk factor for atherosclerosis. Decreased compliance of the endothelial substratum, e.g., vessel stiffness, has also been suggested to lead to EC retraction, whereby the cells react by pulling away from each other [23,24]. It is also associated with increased permeability [25] and/or the decreased expression of adhesion proteins to the luminal side of the endothelium.

In this study, we hypothesized that the endothelium in HAS-2 transgenic mice is affected by increased stiffness in the underlying tunica media. To address this hypothesis, we investigated EC dysfunction via myograph studies, NO bioavailability via diaminofluorescence (DAF), and superoxide formation via dihydroethidium (DHE) fluorescence. To further investigate impaired endothelium-dependent relaxation, we performed immunoblotting to assess ecSOD, eNOS, and eNOS-Ser^1177^ phosphorylation. Moreover, we investigated the distances between ECs via stereological methods.

## 2. Results

### 2.1. Functional Studies and NO Measurements

To examine the contractile responses in the aortas of mice, they were exposed to a depolarizing physiological saline solution with high K+ concentrations (KPSS), and a concentration–response curve was constructed for phenylephrine (PhE). The aortas of wild-type (WT) and HAS-2 mice contracted similarly in response to PhE and KPPS (Figure 1, Table 1, and Figure A1). However, the contractions in the aortas of female mice were increased compared to those of male mice (Figure 1). In the presence of an inhibitor of NO synthase, PhE contractions were markedly increased in the aortas of both WT and HAS-2 mice, and were similar between sexes. 

Concentration–response curves were constructed for ACh and the NO donor, sodium nitroprusside (SNP), to examine endothelium-dependent and -independent relaxations. In the presence of an inhibitor of NOS (L-N^G^-nitroarginine (L-NOARG)), ACh relaxations were abolished (Figure 2a), while the concentration–response curves for SNP were shifted leftward (Figure 2b). In the PhE-contracted aortas of WT mice, ACh and SNP induced concentration-dependent relaxations, which were more potent in preparations from female than male mice (Figure 2). Sex differences in response to ACh and SNP were also present in the aortas of HAS-2 mice (Table 1).

ACh-induced relaxations were blunted in the aortas of male HAS-2 mice compared to those of WT mice, but not in the aortas of female HAS-2 mice (Figure 3a–d). This difference in male aortas was eliminated in the presence of L-NOARG (Figure A2), suggesting a dependence on NO. Additionally, relaxations induced by SNP were reduced in the aortas of male and female HAS-2 versus WT mice (Figure 3e,f). The fact that blunted ACh relaxations occurred in male mice but not in females, despite the similar effects of HAS-2 overexpression on SNP relaxations, suggests that EC dysfunction occurred in the aortas of male HAS-2 mice.

To investigate whether decreased NO levels underlie impaired endothelium-dependent relaxation, they were measured using a NO-sensitive fluorescent dye, DAF, at baseline and after the addition of 10^−^^7^ PhE, followed by 10^−^^5^ M ACh. The baseline NO levels were unchanged in the aortas of male and female HAS-2 versus WT mice (Figure 4a,b). We observed a decrease in fluorescence and, thus, the NO output from the HAS-2 male aortas compared to the WT male aortas after PhE and ACh stimulation (Figure 4a). We detected no differences between the aortas of WT and HAS-2 female mice (Figure 4b).

### 2.2. eNOS Expression and Phosphorylation

To investigate whether impaired endothelium-dependent relaxation and decreased NO output is associated with changes in the NO pathway, we analyzed eNOS protein expression in the WT and HAS-2 aortas and found no change in total eNOS expression. Surprisingly, the aortas of male WT mice displayed significantly lower eNOS expression than those of female WT mice (Figure 5a). 

However, NO formation depends on EC Ca^2+^ and the phosphorylation of eNOS [26]. Therefore, we measured the activation of eNOS via phosphorylation of the Ser^1177^ residue. We found an overall sex-dependent difference, with the ratio significantly higher in male mice than in females (Figure 5b). Moreover, the ratio of P-eNOS (ser^1177^) to total eNOS was significantly upregulated in male HAS-2 mice compared to WT mice. The decreased eNOS expression but increased eNOS-Ser^1177^ phosphorylation in the aortas of male mice may explain the sex differences in NO output in the aortas of WT mice; however, it cannot explain the reduced NO bioavailability in the aortas of HAS-2 male mice compared WT mice. 

The eNOS dimer produces NO, and its monomer produces O_2_-; therefore, the dimer/monomer ratio is an expression of eNOS-dependent NO/O_2_- balance. The dimer/monomer ratio was higher in male aortas than in those of female mice (Figure 5c),and did not differ between the aortas of WT and HAS-2 mice. Therefore, the uncoupling of eNOS does not explain the impaired ACh-induced NO-dependent relaxations in the aortas of HAS-2 male mice.

### 2.3. Assessment of Vascular O_2_- and O_2_- Scavenging

eNOS is not the only source of superoxide formation in the vascular wall. Therefore, we measured total superoxide using DHE, which is a cell-permeable dye that emits blue fluorescence. In the presence of O_2_-, DHE is oxidized to ethidium, which is impermeable to intact cell membranes and emits red fluorescence, and was detected in sections from both HAS-2 and WT animals; however, it did not occur in sections treated with polyethylene glycol-conjugated SOD (PEG-SOD) (Figure 6a–f). The relative fluorescence intensity was markedly increased in the aortas of HAS-2 animals of both sexes compared to WT animals (Figure 6f), but there were no differences in the aortas of male and female mice.

To gain further insight, we investigated the main scavenger of superoxide in the vasculature: ecSOD. We found a statistically significant decrease in ecSOD expression in the HAS-2 animals compared to the WT animals (Figure 7). There were no sex-dependent differences. The downregulation of ecSOD and increase in superoxide may have contributed to the reduced bioavailability of NO and blunted ACh relaxations in the aortas of male HAS-2 mice. The lack of change in response in the aortas of female HAS-2 mice could be ascribed to the requirement for less NO to induce relaxations of similar magnitude in the aortas of female mice, as evidenced by their higher sensitivity to SNP.

### 2.4. EC–EC Distance and Expression of Occludin, Vascular Endothelial Cadherin, and the LDL Receptor

An alternative explanation for changed EC function and Ca^2+^ levels is that the underlying hyaluronan accumulation alters the distances, and possibly the signaling, between ECs. Therefore, we investigated the distances between ECs in the intima of the aortas using 56,000×-magnified images of two adjacent ECs (Figure 8a). An average of 325 measurements were taken of each aorta, and the mean of these measurements showed that the average distance between ECs in the HAS-2 mice was 2 nm broader than in the WT mice (Figure 8b).

To further investigate this difference, we determined the distribution of the measurements and found no significant difference between WT and HAS-2 mice following a two-way ANOVA. However, using a t-test, we observed a lower percentage of distances of 20–25 nm and a higher percentage of distances of 25–40 nm in male HAS-2 mice versus WT mice (Figure 8d).

To further explore the significance of the increased EC–EC distance and dysfunction, we examined the adherence and tight junction proteins, as well as occludin and vascular endothelial cadherin (VE-cadherin) expression. Immunoblotting did not show any genotypic or sex-related differences in the expression of occludin and VE-cadherin (Figure 9a,b). 

Hyaluronan overexpression may increase the vascular wall’s susceptibility to atherosclerosis development. We therefore examined whether there was altered expression of low-density lipoprotein (LDL) receptors. LDL receptor expression was increased in the aorta of HAS-2 mice compared to those of WT mice, but a post hoc Šídák’s multiple comparisons test revealed that this result was driven mainly by the differences in male, but not female, mice (Figure 9c). Transglutaminase 2 (TG2) cross-links extracellular matrix proteins and has a complex role in atherosclerosis, participating in the initial stages of plaque formation [27], while its deficiency could be associated with unstable atherosclerotic plaques [28] in mice. Immunoblotting showed significantly lower expression of TG2 in the aortas of female versus male HAS- 2 mice and a significant decrease in TG2 in HAS-2 female mice compared to WT mice (Figure A3). Taken together, these results suggest upregulation of the mechanisms that could increase atherosclerotic plaque formation in the aorta walls of male HAS-2 mice.

## 3. Discussion

The novel findings of the present study are that hyaluronan increases oxidative stress in the vascular wall and that together with increased EC distance, it is associated with a sex-specific decrease in NO levels and endothelial dysfunction in the aorta of male HAS-2 transgenic mice. Meanwhile, dysfunction of the smooth muscle, characterized by relaxations induced by the NO donor SNP, seems to affect HAS-2 mice independently of sex. Additionally, in the wild-type mice, the concentration–response curves for ACh and SNP are rightward shifted in male mice, suggesting that higher concentrations of NO are required to obtain similar relaxations to those in the aortas of female mice. This difference may explain why endothelial function is more susceptible to changes in male HAS-2 transgenic mice, despite the occurrence of higher levels of eNOS-Ser^1177^ phosphorylation, a higher eNOS dimer/monomer ratio, and similar oxidative stress levels in the aortas of male and female HAS-2 transgenic mice. 

### 3.1. Sex-Dependent Differences in Endothelial Dysfunction

Sex hormones affect the cardiovascular system, and comparisons of arteries from male and female animals often show that estrogens are protective and improve endothelial function [29]. Sex-specific differences have been observed whereby the aortas of female ICR (Institute of Cancer Research) mice react more promptly to ACh than those of male ICR mice, suggesting better endothelial function in female ICR mice [30,31]. A recent study showed that a sex-specific contractile prostanoid contributes to lower ACh relaxation in the mesenteric arteries of male mice [32]. Moreover, contractile prostanoids may account for the sex-specific differences in spontaneously hypertensive rats [33]. In the present study, ACh induced more potent relaxations in the aortas of female versus male wild-type mice. The presence of a NO synthase inhibitor led to the inhibition of ACh relaxations, and increased tension to a larger degree in the aortas of male mice than those of females; this suggests that the more potent relaxations in the female aorta can be attributed both to differences in the NO signaling pathway and the lower contribution of endothelium-derived contractile factors. Several other findings support this observation; at the level of ECs, eNOS expression is decreased, with increased phosphorylation of eNOS-Ser^1177^ and an increased eNOS dimer/monomer ratio, leading to higher levels of NO in the aortas of male versus female wild-type mice. At the smooth muscle layer level, SNP induces more pronounced relaxations in the aortas of female versus male wild-type mice, suggesting that higher NO levels are required to obtain comparable relaxations in the aortas of male mice. 

Premenopausal women tend to experience cardiovascular disease less frequently than age-matched males, and this sex-dependent difference in cardiovascular risk also appears in people with diabetes [34]; endothelium-dependent vasodilatation may partly explain this difference [29,35]. In contrast, we found that the impairment of endothelium-dependent relaxation in the aortas of female diabetic db/db mice was more pronounced than in age-matched male mice [36]; however, these type 2 diabetic mice, in addition to hyperglycemia, also have a leptin receptor defect that contributes to endothelial dysfunction [37]. Therefore, the finding of a sex-specific effect of endothelium-dependent relaxation on the aortas of male HAS-2 transgenic mice suggests that these mice could serve as a suitable experimental model for exploring diabetic macroangiopathy, with characteristics similar to those occurring in patients. 

### 3.2. Effect of Hyaluronan Overexpression on Endothelial Cell Function 

Several mechanisms, including changes in eNOS uncoupling [17], Akt/eNOS phosphorylation [3,31], EC Ca^2+^ signaling [38], and increased oxidative stress [17], have been suggested to play a role in EC dysfunction in diabetes. Investigating eNOS expression, eNOS-Ser^1177^ phosphorylation, and the eNOS dimer/monomer ratio did not reveal differences in the aortas of HAS-2 transgenic mice versus wild-type mice. These findings suggest that additional cardiovascular risk factors, e.g., high plasma glucose, hyperlipidemia, and hypertension, may have to be present to induce changes in the NO pathway. 

Damage induced by oxidative stress increases significantly when EcSOD activity is decreased [20]. EcSOD also binds glycosaminoglycans and prevents the oxidative degradation of hyaluronan [21]. In the present study, oxidative stress, measured via DHE staining, was markedly increased in the aortas of HAS-2 transgenic mice, which can be explained by a decrease in EcSOD expression. The oxidative stress levels were similar in the aortas of male and female HAS-2 transgenic mice. However, the increase in superoxide may be sufficient to explain the decreased bioavailability of NO. Thus, blunted endothelium-dependent relaxation may only affect the aortas of male HAS-2 mice, since higher NO levels are required to obtain relaxations comparable to those in the aortas of female mice.

The aortas of HAS-2 transgenic mice have increased stiffness and a changed smooth muscle phenotype [10,11]. Arterial stiffness is associated with inflammation, oxidative stress, smooth muscle proliferation, migration, and endothelial dysfunction [25]. Moreover, it has been postulated that ECs in a confluent monolayer on a stiffer substrate disrupt cell–cell contacts and increase permeability [24]. Recently, we found that mRNA expression of the three classical adhesion molecules, E-selectin, V-CAM, and I-CAM, was similar in the aortas of HAS-2 transgenic and control mice. These results suggest that increased HA accumulation in the medium does not necessarily affect the EC layer [11], although further studies are required to clarify whether there are changes in the EC cytoskeleton. However, our measurements of the distances between ECs showed increased distances in the aortas of HAS-2 transgenic mice. Gap junctions transmit membrane potentials and Ca^2+^ levels in the EC monolayer [39]. We can only speculate that the increased EC–EC distances may be associated with lower EC Ca^2+^ levels, and that they contribute to decreased NO formation and endothelial dysfunction in the aortas of male HAS-2 transgenic mice.

### 3.3. Pro-Atherogenic Effects of Hyaluronan Overexpression

In a previous study, hyaluronan overexpression in the tunica media of HAS-2 transgenic mice was found to promote the development of atherosclerosis in ApoE^−/−^ mice [10]. A limitation of this study was that it did not address sex differences. However, in LDL receptor-deficient mice, the incidence of atherosclerotic lesions in the aorta was significantly higher in males than in females, and the same tendency was found in the aortas of ApoE^−/−^ mice [40]. Although male-specific endothelial dysfunction in the current study could explain the increased atherosclerosis in HAS2 transgenic mice, further investigations using other approaches are required to address whether EC dysfunction accelerates atherosclerosis in HAS-2 transgenic mice.

EC injury and increased permeability are thought to play a role in atherogenesis [41]. In this study, the distances between ECs were increased, but this was not associated with changes in VE-cadherin and occludin in the aortas of HAS-2 transgenic mice. Therefore, the functional effects of an increase in EC distance on atherogenesis are unclear. 

The subendothelial retention of atherogenic lipoproteins is now considered central to the development of atherosclerotic lesions [8,42]. There is also evidence that the subendothelial environment changes, e.g., lipoproteins may bind directly to the proteoglycans [5], and even more to hyaluronan cable structures that are formed in the aortas of HAS-2 transgenic mice [11]. TG2, which cross-links peptidoglycans, has been suggested to play a deleterious role in early atherogenesis [27]. However, TG2 is considered protective as it stabilizes more advanced atherosclerotic lesions in ApoE^−/−^ mice [28]. Moreover, recent studies by our group revealed that increased TG2 activity was linked to reduced sensitivity to NO in the smooth muscle of small mesenteric arteries [43]. In the present study, TG2 is less expressed in the aortas of female HAS-2 transgenic mice than in those of their male counterparts, suggesting that decreased TG2 could play a role in the vasoprotection that occurs in female mice.

### 3.4. Limitations

One of the limitations of this study is the chosen mouse model, in which we can only observe how the excessive accumulation of hyaluronan and increased aortic stiffness affects the endothelium, without other influences such as hyperglycemia or age, which is usually the case. This, however, is also the strength of using this mouse model because it gives us the unique possibility of investigating how hyaluronan accumulation and arterial stiffness affect the endothelium without the interference of hyperglycemia and hyperlipidemia.

Moreover, the present study does not address whether the pro-atherogenic environment reflected by increased stiffness [10], changes in the vascular smooth muscle phenotype [11], endothelial cell dysfunction, and the upregulation of LDL receptors is associated with increased susceptibility to atherosclerosis development. In a previous study, we observed more pronounced atherosclerosis in the aorta of the HAS-2 transgenic mice [10]. However, other approaches will be required to measure whether the changes in the pro-atherogenic environment are indeed associated with a larger-scale transport of atherogenic lipoproteins into the vascular wall.

The increased distance between ECs may also contribute to EC dysfunction in the aortas of male HAS-2 mice by affecting signaling in the ECs. We have previously observed in small mesenteric arteries from type-2 diabetic db/db mice that impaired endothelium-dependent relaxations are associated with decreased endothelial calcium levels [38], and we cannot exclude that contribute to the impaired endothelium-dependent relaxation in the aorta of male HAS-2 transgenic mice. 

## 4. Materials and Methods

### 4.1. Animals

Unless otherwise stated, we used 4–5-month-old age- and sex-matched C57BL/6J mice overexpressing HAS-2, and WT mice [10]. The mice were housed at the animal facility at Aarhus University and handled according to the guidelines and procedures recommended by The Animal Experiments Inspectorate, Denmark. They were kept at 21 °C and were given free access to standard chow and water. The mice were kept under a 12 h light/dark cycle, and to avoid the influence of torpor [44,45], they underwent experimentation early in the light period. 

### 4.2. Contractility

Thoracic aorta segments measuring 2 mm in length from WT and HAS-2 mice were mounted on parallel 100 µm wires in double myographs, and bathed in physiological saline solution (PSS) comprising: 119 mM NaCl, 4.7 mM KCl, 1.17 mM MgSO_4_, 25 mM NaHCO_3_, 1.18 mM KH_2_PO_4_, 5.5 mM glucose, 1.6 mM CaCl_2,_ and 0.026 mM EDTA. The PSS was bubbled with 5% CO_2_ in air at 37 °C [46]. After an equilibration period, the vessel diameter was adjusted to obtain maximal active tension development [46]. To check vessel viability, a standard start was performed involving two instances of stimulation using a solution with the same composition as the PSS, except that NaCl was exchanged for KCl on an equimolar basis (KPSS, 123 × 10^−3^ M). Two myographs were incubated for 30 min: one without an inhibitor of NOS, L-NOARG (3 × 10^−4^ M), and one with. Concentration–response curves were constructed as follows: (1) PhE was added in increasing concentrations. (2) The vessels were contracted using 10^−7^ M PhE, and then, relaxed using increasing concentrations of ACh. (3) The vessels were contracted using 10^−7^ M PhE, and then, relaxed using increasing concentrations of SNP. 

### 4.3. Diaminofluorescence

Fresh aorta segments were incubated in a Krebs-HEPES buffer (119 mM NaCl, 20 mM HEPES, 1.2 mM CaCl_2_, 4.6 mM KCl, 1 mM MgSO_4_, 0.4 KH_2_PO_4_, 5 mM NaHCO_3_, 5.5 mM glucose, and 0.15 Na_2_HPO_4_, at pH 7.4) at 37 °C for 30 min. Subsequently, the aortas were incubated in 300 µL Krebs-HEPES with 2 × 10^−5^ M 4.5-diaminofluorescein (DAF-2) at 37 °C for 45 min. For the negative controls, a segment of WT and a segment of HAS-2 aorta were incubated with a NOS inhibitor, L-N^G^-nitroarginine methyl ester (L-NAME, 10^−4^ M). A tube without an aorta was included as a blank.

The fluorescence was measured using a fluorometer in a black 96-well plate with a clear bottom. The aortas and 200 µL of the medium were added to each well, along with 0.05% triton to permeabilize the aorta, followed by a 5 min stabilizing period (in the dark at 37 °C). The aortas were removed, the samples were excited with light with a wavelength of 495 nm, and the basal fluorescence was measured at 515 nm. The aortas were returned to the well with 10^−7^ M PhE for 5 min of incubation (dark, 37 °C), and the fluorescence was measured. This was repeated with 10^−5^ M ACh. The concentrations were chosen on the basis of the results from the contractility experiments. The aortas were dried for 5 min on tissue paper, and their weights were registered to normalize the NO release by weight.

### 4.4. Detection of Vascular O_2_− 

Vascular O_2_− was detected using the oxidative fluorescent dye dihydroethidium (DHE, Molecular Probes, Eugene, OR, USA) [20,47]. Fresh aorta segments were frozen in optimum cutting temperature compound immediately after excision. Frozen sections were cut to a length of 10 µm and were treated with DHE (4 × 10^−6^ M concentration; 200-μL volume per section) or with PBS (200 μL volume; vehicle/time control). All sections were incubated at 37 °C in a light-protected, humidified chamber for 30 min, and rinsed once with 400 μL of PBS to remove unoxidized DHE. Some sections were preincubated with PEG-SOD (100 U ml−1) as a negative control. O_2_- production was estimated using confocal microscopy (Zeiss LSM 510 Meta; Carl Zeiss, Oberkochen, Germany). The settings for laser scan imaging included a 512 × 512 pixel resolution; argon/krypton laser power: 6%; objective: ×40 NA 1.2; ethidium bromide (EtBr) excitation: 488 nm; and EtBr emission: 580 to 630 nm band-pass filter. Identical photomultiplier tube voltage (725 V) and gain (2.0) settings were used for all sections. At least two different sections from each vascular ring were used, and four fields of view from each vascular ring were imaged such that no regional overlap occurred. Maximum-intensity z-projections were used to quantify relative fluorescence intensity (RFI) using MetaMorph Image Analysis Software (Universal Imaging, Molecular Devices Corp., Downingtown, PA, USA). The data are reported as RFI units from 0 (least intensity) to 255 (greatest intensity). This unit was arbitrarily defined by the digitizer gain, which was held constant for all samples.

### 4.5. Immunoblotting

The heart and the entire aorta were removed from each mouse. The aorta was separated from the heart and the adventitia was gently cut off. Two aortas were pooled, and the tissues were homogenized in a 300 µL sample buffer (0.1% SDS, 1% NP-40, 0.5% sodium deoxycholate, 100 × 10^−3^ M NaCl, 10^−4^ M sodium orthovanadate, 10^−3^ M sodium fluoride, and 5 × 10^−2^ M Tris-HCl, pH 7.4) containing a mixture of protease inhibitors (10 µg/mL aprotinin, 10 µg/mL leupeptin, 5 µg/mL pepstatin A, and 10 µg/mL PMSF). Samples were centrifuged to form pellet cell debris. The protein concentration was measured using a MicroBCATM Protein Assay Reagent Kit (Pierce, Rockford, IL, USA). 

Equal amounts of total protein (2 µg for EcSOD detection, 20 µg for the detection of other proteins of interest) were resolved via electrophoresis on 4–20% Criterion™ TGX Stain-Free™ SDS-polyacrylamide Gel (Bio-rad, Hercules, California) under reducing conditions. Before the transfer, the gel was activated by UV light for 5 min. After the transfer, the PVDF membrane (PerkinElmer, Waltham, MA, USA) was reactivated by UV light, and the total amount of protein was visualized for loading control and normalization [48]. The membrane was blocked overnight at 4 °C in 5% non-fat powdered milk, washed, and incubated with primary antibody (anti-EcSOD 1:10.000: BioVision Research Products, Milpitas, California; eNOS 1:1000, #5589: Abcam, Cambridge, UK; P-eNOS 1:1000, #9571S: Cell Signaling Technologies (CST), Boston, MA, USA; occludin 1:1000, #EPR20992: Abcam, Cambridge, UK; VE-cadherin 1:1000: Abcam, Cambridge, UK; LDL receptor 1:500, #EP1553Y: Abcam, Cambridge, UK; and TG2 1:200, ab421: Abcam, Cambridge, UK). This was followed by incubation with a relevant streptavidin–horseradish peroxidase (HRP)-labeled secondary antibody (1:30,000) (Santa Cruz, Dallas, TX, USA). 

The blot was developed using Western LightningTM Chemiluminescence Reagent Plus (PerkinElmer, Waltham, MA, USA). For all immunoblotting experiments, samples from HAS-2 mice were separated on the same gel as those of their WT littermates. An internal control was included on all membranes to adjust for inter-membrane differences. Some membranes were stripped with a 6M GnHCl solution for 10 min, blocked, and then, re-incubated with a new antibody. The results were normalized to the total protein transferred.

### 4.6. EC–EC Distance

Already-sampled aortas were used to measure the EC–EC distance via electron microscopy [10]. New images of 50 nm sections were taken at 40 evenly distributed spots on the aorta, at a magnification of 56,000×, using an FEI Morgani electron microscope with a SIS MegaView III digital camera and iTEM software (Olympus Soft Imaging Solutions GmbH, Münster, Germany). The distance between the cell membranes of adjacent ECs was measured in iTEM, and test lines spaced 150 × 150 nm apart determined where the measurements were made. The distribution of the measurements was determined as a percentage of the total number of measurements for each aorta.

### 4.7. Data and Statistical Analyses

The myograph data were recorded using LabChart (ADInstruments, Dunedin, New Zealand). The Western blot data were recorded using Image Lab (Bio-Rad, Hercules, California). The data were processed in Microsoft Office Excel, and data and graph analyses were performed in GraphPad Prism 8 software (GraphPad Prism, San Diego, CA, USA, RRID: SCR_002798). Outlier calculations were performed using the Grubs test, and one maximal data point was excluded from the data set (GraphPad Prism). When analyzing only two groups, the statistical comparison was performed using the Student’s *t*-test or the Mann–Whitney rank sum test. Statistical analysis of grouped data was performed using a two-way ANOVA. The models’ assumptions were investigated by inspecting Q-Q plots, and the data were logarithmically transformed when necessary to generate a Gaussian-distributed data set. *p* < 0.05 was considered statistically significant. Post-tests were only performed if F was significant and there was no variance in inhomogeneity.

## 5. Conclusions and Perspective

In previous studies, hyaluronan has been associated with increased stiffness and accelerated atherosclerosis in HAS-2 transgenic mice [10,11]. The findings of the present study suggest that hyaluronan increases oxidative stress in the vascular wall, and that together with increased EC distance, it is associated with a sex-specific decrease in NO levels and endothelial dysfunction. Diabetes-related hyaluronan accumulation, followed by oxidative stress and the upregulation of LDL receptors, leads to increased susceptibility of the vascular wall to atherosclerosis, a process that can be further accelerated by EC dysfunction. 

## Figures and Tables

**Figure 1 ijms-24-08436-f001:**
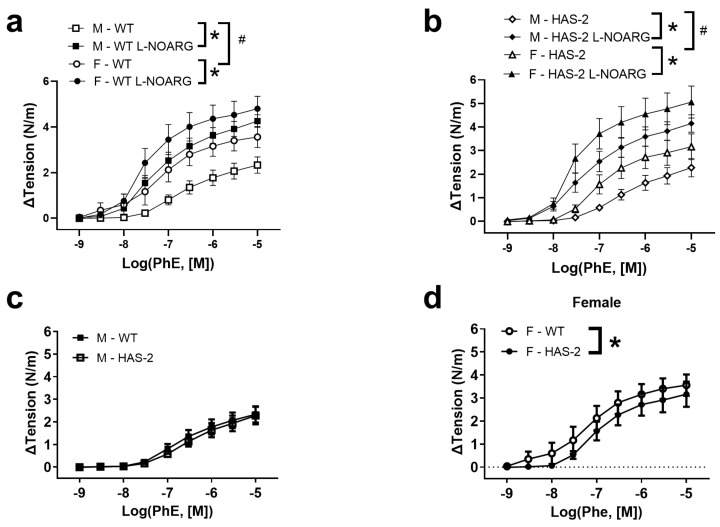
Response of aortas from WT and HAS-2 mice to phenylephrine, presented as mN contraction per mm aorta. Average contractions measured as an increase in tension (N/m) induced in the aortas of (**a**) male (n = 13/group) and (**b**) female (n = 5/group) wild-type (WT) mice and hyaluronan synthase (HAS-2) transgenic mice. (**c**) Graph of the contraction response of male WT and HAS-2 aortas. We analyzed the data via two-way ANOVA, and we detected no significant interactions or differences between the WT and the HAS-2 aortas (*p* = 0.47, n = 13/group). (**d**) Graph of the contraction response of female WT and HAS-2 aortas to phenylephrine. We analyzed the data via two-way ANOVA and detected no significant interaction, but the HAS-2 aortas contracted slightly less than the WT aortas (*p* = 0.025, n = 5/group). Results are presented as the means ± s.e. * *p* < 0.05 versus the control curve. # *p* < 0.05 versus the response in the aortas of female mice (two-way ANOVA).

**Figure 2 ijms-24-08436-f002:**
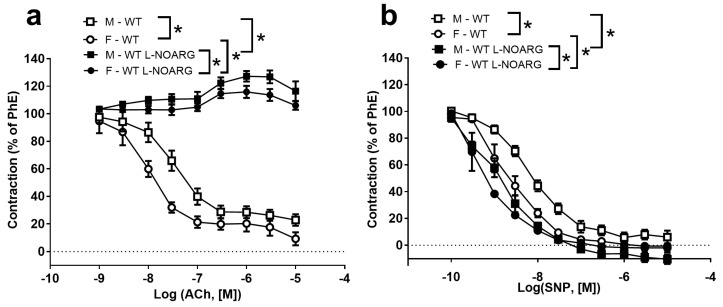
Sex-specific differences in acetylcholine and sodium nitroprusside relaxation in the aortas of wild-type (C57Bl6J) mice. Average concentration–response curves for (**a**) acetylcholine (ACh) and (**b**) sodium nitroprusside (SNP) in phenylephrine (PhE)-contracted aortas of male (M) and female (F) mice. The curves were obtained in the absence and the presence of an inhibitor of nitric oxide (NO) synthase (L-NOARG) (3 × 10^−4^ M). Results are presented as the means ± s.e. * *p* < 0.05 according to two-way ANOVA.

**Figure 3 ijms-24-08436-f003:**
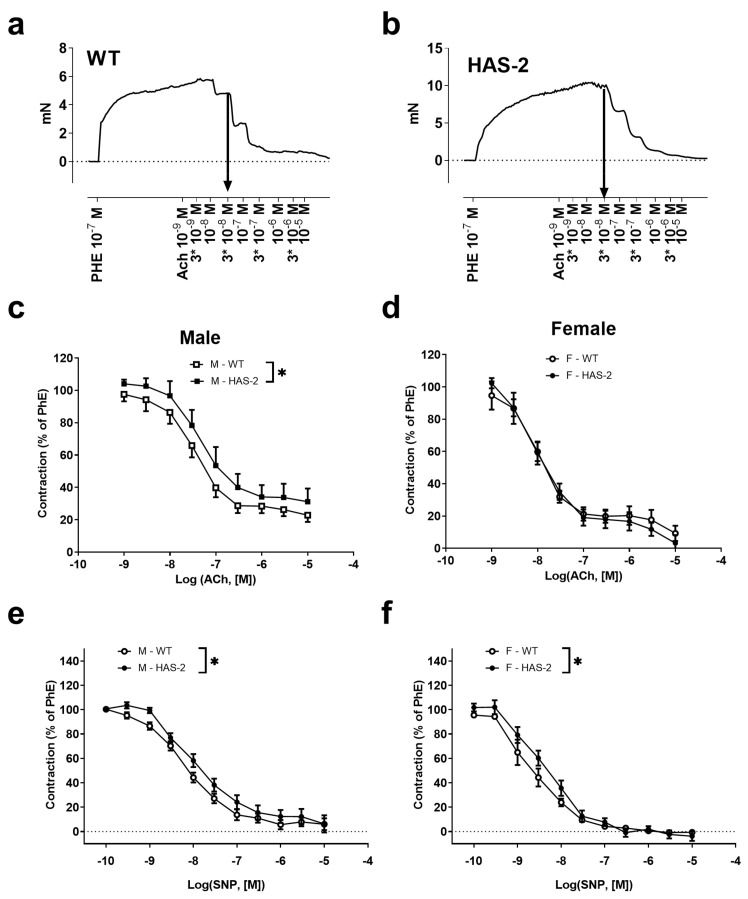
Blunted acetylcholine and sodium nitroprusside relaxation in the aortas of male mice overexpressing hyaluronan synthase 2 (HAS-2). (**a**,**b**) Traces representative of relaxation induced by increasing concentrations of acetylcholine (ACh) in the aortas of wild-type (WT) (**a**) and HAS-2 (**b**) male mice. (**c**,**d**) Averaged concentration–response curves for ACh in WT and HAS-2 male (**c**) (n = 11) and female (**d**) (n = 5) mice (n = 11). (**e**,**f**) Averaged concentration–response curves for sodium nitroprusside (SNP) in aorta segments from male (**e**) (n = 11) and female (**f**) (n = 5) mice. The results are presented as the means ± s.e. * *p* < 0.05 according to two-way ANOVA.

**Figure 4 ijms-24-08436-f004:**
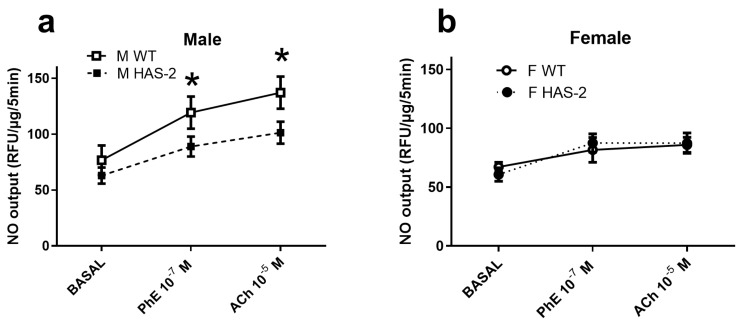
Nitric oxide (NO) levels were decreased in the aortas of male mice overexpressing hyaluronan synthase 2 (HAS-2). Average diaminofluorescence (DAF) in aortas of male (**a**) (n = 7–15) and female (**b**) (n = 7) mice measured at baseline level (BASAL) and in response to phenylephrine (PhE) and acetylcholine (ACh), and normalized to tissue weight. The results are presented as the means ± s.e. * *p* < 0.05 compared to the aortas of WT mice according to two-way ANOVA.

**Figure 5 ijms-24-08436-f005:**
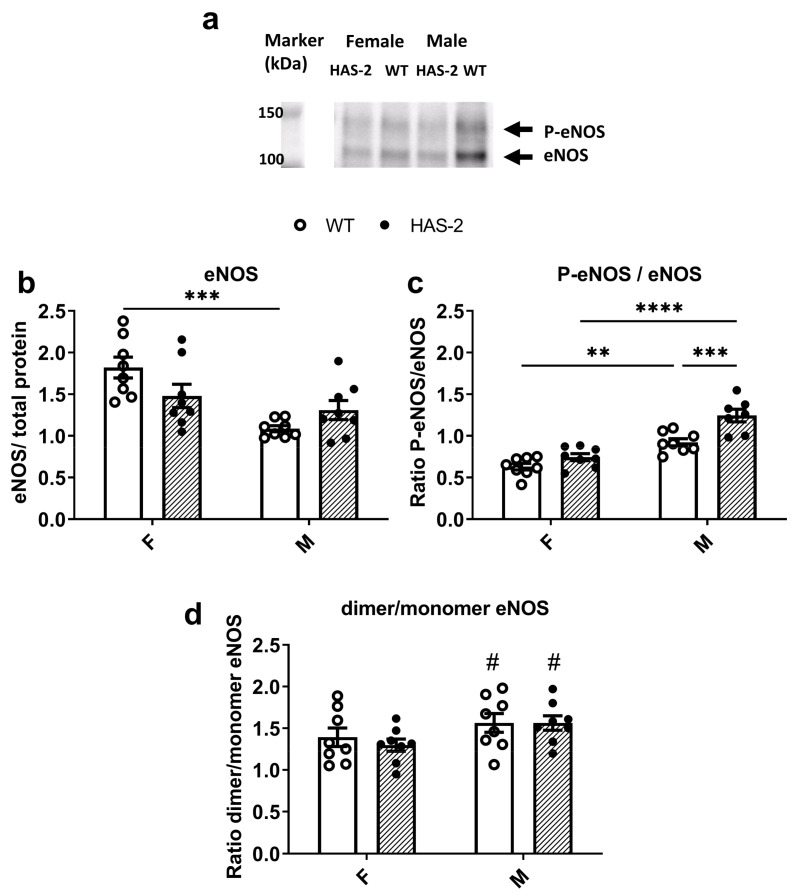
Expression of endothelial nitric oxide synthase (eNOS) and eNOS-Ser^1177^ phosphorylation (P-eNOS). (**a**) Representative immunoblot of P-eNOS, located at approximately 150 kDa, and eNOS, located at approximately 130-140 kDa, in aortas of wild-type (WT) and mice overexpressing hyaluronan synthase 2 (HAS-2). (**b**) Average eNOS expression in WT and HAS-2 female (F) and male (M) mice. (**c**) Average p-eNOS/eNOS ratio in HAS-2 mice and their WT littermates. (**d**) Average dimer/monomer eNOS ratio in HAS-2 and WT mice. ** *p* < 0.01, *** *p* < 0.001, and **** *p* < 0.0001 according to two-way ANOVA with Šídák’s multiple comparisons test. # *p* < 0.05 compared to female mice according to two-way ANOVA.

**Figure 6 ijms-24-08436-f006:**
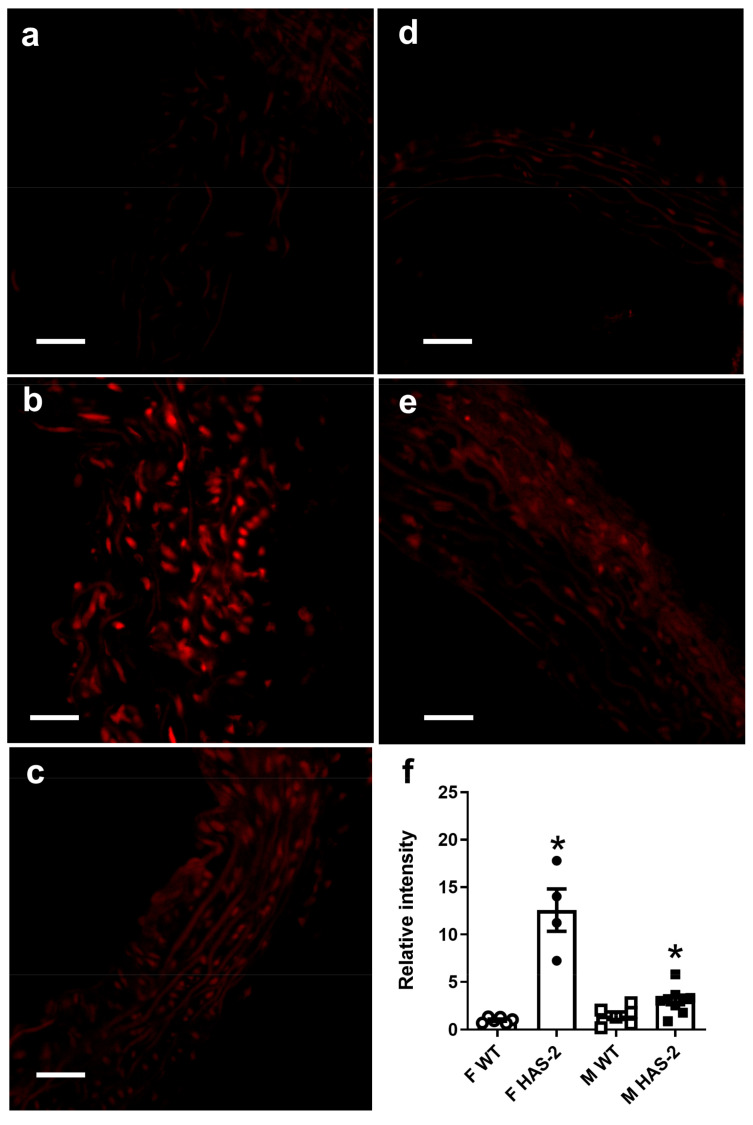
HAS-2 overexpression increases O_2_- production in the aorta. Representative vessel from each group showing dihydroethidium (DHE) staining for O_2_− production. (**a**) Female wild-type (WT) mice. (**b**) Female transgenic mice overexpressing hyaluronan synthase 2 (HAS-2). (**c**) DHE staining after polyethylene glycol-conjugated superoxide dismutase (PEG-SOD) pretreatment of the aortas of female HAS-2 transgenic mice. (**d**) Male WT mice. (**e**) male HAS-2 transgenic mice. Calibration bar: 100 μm. (**f**) Average values for DHE staining in the aortas are presented as the means ± s.e. Comparisons were made according to two-way ANOVA. * *p* < 0.05 versus wild-type aorta.

**Figure 7 ijms-24-08436-f007:**
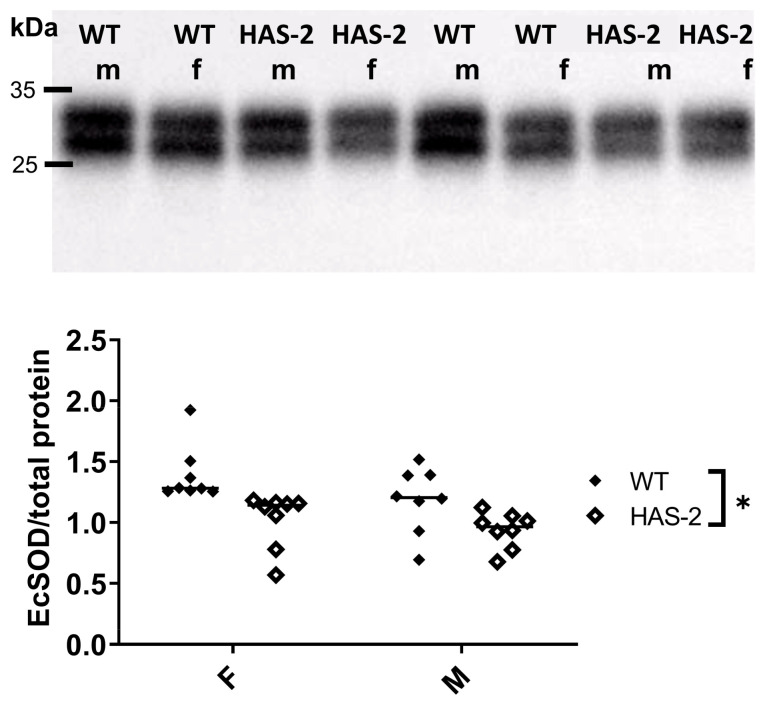
Extracellular superoxide dismutase (EcSOD) is downregulated in the aortas of mice overexpressing hyaluronan synthase 2 (HAS-2). Representative section of an ecSOD Western blot and the corresponding average results normalized to total protein. * *p* < 0.05 according to two-way ANOVA.

**Figure 8 ijms-24-08436-f008:**
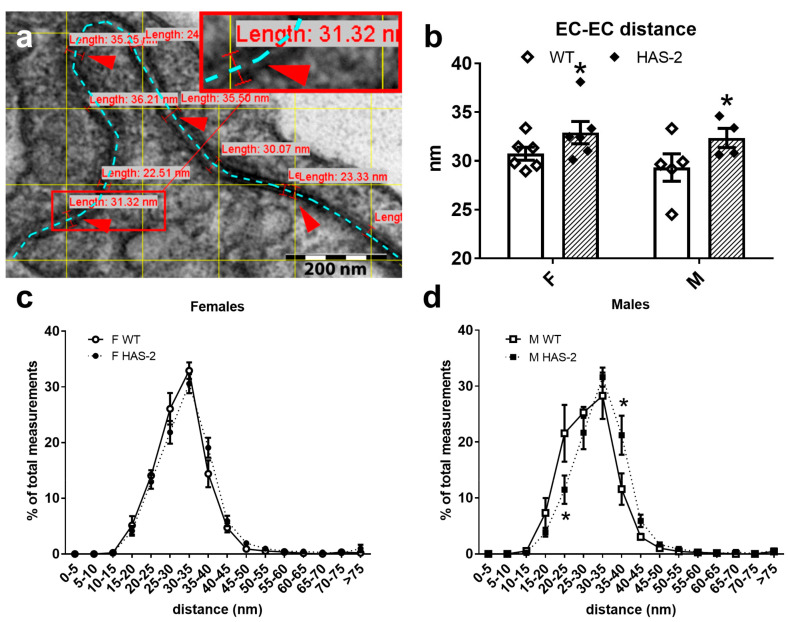
Endothelial cell–cell distance increases in HAS-2 mice. (**a**) Electron micrograph image showing how distance measurements were made. The blue dotted line indicates the border between endothelial cells (ECs). Red arrowheads indicate some of the spots where distance was measured. The red square is a magnification of one of those spots. (**b**) Average distance between ECs from wild-type mice (WT) and transgenic mice overexpressing hyaluronan synthase 2 (HAS-2). Each point represents the mean of ~329 measurements. * *p* < 0.05 according to two-way ANOVA. Graphs depict the distribution of measurements in the WT vs. HAS-2 females (**c**) and males (**d**). * *p* < 0.05 at the indicated distances compared WT.

**Figure 9 ijms-24-08436-f009:**
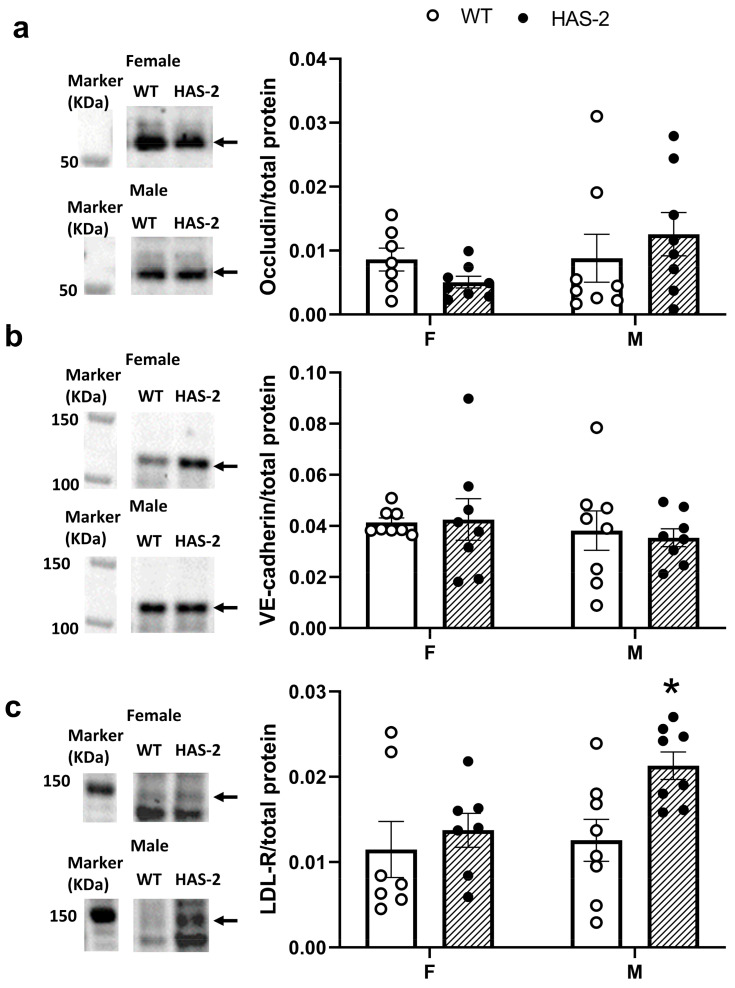
Male mice overexpressing hyaluronan synthase 2 (HAS-2) showed increased expression of the low-density lipoprotein receptor (LDL-R). Representative immunoblots (left) and average protein expression of (**a**) occludin, (**b**) vascular endothelial cadherin (VE cadherin), and (**c**) LDL-R. Immunoblotting for occludin in the aortas of male (M) and female (F) wild-type transgenic mice and mice overexpressing HAS-2. Arrows indicate the quantified band, which in the case of LDL-R, is based on the band for the positive control. The data are presented as the means ± s.e. * *p* < 0.05 versus WT mice according to two-way ANOVA with Šídák’s multiple comparisons tests.

**Table 1 ijms-24-08436-t001:** Sensitivity of responses induced by phenylephrine (PhE), acetylcholine (ACh), and sodium nitroprusside (SNP) in the aortas of wild-type (WT) and hyaluronan synthase (HAS-2) transgenic male and female mice.

	Male WT	Male HAS-2	Female WT	Female HAS-2
PhE −log EC_50_	6.49 ± 0.11 # (13)	6.35 ± 0.11 # (13)	7.19 ± 0.24 (5)	6.92 ± 0.08 (5)
ACh −log EC_50_	7.45 ± 0.20 # (11)	7.26 ± 0.23 # (11)	8.13 ± 0.25 (5)	7.92 ± 0.11 (5)
SNP −log EC_50_	8.05 ± 0.05 # (11)	7.70 ± 0.07 #* (119)	8.62 ± 0.05 # (5)	8.31 ± 0.05 #* (5)

EC_50_ = half-maximal response. Results are presented as the means ± s.e., and (n) = the number of animals. # *p* < 0.05 versus the same parameter in female mice.* *p* < 0.05 versus the response in the aortas of ΔWT mice.

## Data Availability

Not applicable.

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
