# Peer review of "Sex-Dependent Impairment of Endothelium-Dependent Relaxation in Aorta of Mice with Overexpression of Hyaluronan in Tunica Media"

_ijms, 2023, doi:10.3390/ijms24098436_

Round 1

Reviewer 1 Report

The manuscript submitted to the International Journal of Molecular Sciences by Dr. Karen Axelgaard Lorentzen et al. and entitled «Overexpression of hyaluronan in tunica media is associated with endothelial dysfunction in pre-atherosclerotic male mice» is aimed to in vivo study of fundamental mechanisms and pathways involved in the endothelial dysfunction underlying atherosclerosis. Authors suggested that hyaluronan increases oxidative stress in the vascular wall and increased endothelial cell distance followed by sex-specific endothelial dysfunction and LDL receptor upregulation. Authors concluded that diabetes-related hyaluronan accumulation increases the susceptibility of the vascular wall to atherosclerosis. Actually, conclusions made by the authors are not supported by the obtained results due to incorrect methodology for assessing endothelial dysfunction and atherosclerosis. Please refer to my key remarks.

1. While reading the article, it remains unclear how the authors accessed endothelial dysfunction in mice included in the experiment. If the authors draw a conclusion about the development of endothelial dysfunction based on the assessment of eNOS protein expression, this is a fundamentally wrong approach. A number of both in vivo and in vitro studies show that the role of eNOS in the development of endothelial dysfunction is significantly exaggerated (it has been shown that endothelial dysfunction caused by different triggers is associated with increased of many molecular markers of endothelial dysfunction, excepting eNOS, please refer to Shishkova D.K. et al. Calciprotein Particles Link Disturbed Mineral Homeostasis with Cardiovascular Disease by Causing Endothelial Dysfunction and Vascular Inflammation. International Journal of Molecular Sciences, 2021, 22(22):12458, DOI:10.3390/ijms222212458), so it is not correct to use this marker alone. I recommend use the panel of endothelial dysfunction markers proposed by Dr. Kutikhin et al. (Kutikhin A.G. et al. Endothelial dysfunction in the context of blood-brain barrier modeling. Journal of Evolutionary Biochemistry and Physiology, 2022, 58(3):781-806, DOI:10.1134/S0022093022030139) to assess this pathological condition and analyze these markers not only at the proteomic, but also at the genetic (gene expression profiling) level.

2. The similar problems with the assessment of atherosclerosis – authors used TG2 and LDL receptor expression, but it is not marker of atherosclerosis. Authors must perform the Oil Red staining of mice aorta to for quantification of atherosclerosis lessions (Andrés-Manzano M.J. et al. Oil Red O and Hematoxylin and Eosin Staining for Quantification of Atherosclerosis Burden in Mouse Aorta and Aortic Root. Methods Mol Biol, 2015, 1339:85-99, DOI:10.1007/978-1-4939-2929-0_5).  

3. The age of mice at 4-5 months is too young to develop signs of atherosclerosis, which could be experimentally fixed, given that the mice received a normal (not hyperlipidemic) diet. I recommend revising the design of the study, changing the diet, and raising the age of withdrawal of mice from the experiment, which will allow to get more correct results.

4. The quality of blot images is too low (high background, out of focus, fuzzy images). Please improve it.

Author Response

The manuscript submitted to the International Journal of Molecular Sciences by Dr. Karen Axelgaard Lorentzen et al. and entitled «Overexpression of hyaluronan in tunica media is associated with endothelial dysfunction in pre-atherosclerotic male mice» is aimed to in vivo study of fundamental mechanisms and pathways involved in the endothelial dysfunction underlying atherosclerosis. Authors suggested that hyaluronan increases oxidative stress in the vascular wall and increased endothelial cell distance followed by sex-specific endothelial dysfunction and LDL receptor upregulation. Authors concluded that diabetes-related hyaluronan accumulation increases the susceptibility of the vascular wall to atherosclerosis. Actually, conclusions made by the authors are not supported by the obtained results due to incorrect methodology for assessing endothelial dysfunction and atherosclerosis. Please refer to my key remarks.

  • We have in the manuscript reworded several sentences, hopefully increasing the understanding of the background for the study. We thank the reviewer for the comments and suggestions. In a previous study, we found that HAS-2 upregulation accelerates the development of atherosclerosis in ApoE knockout mice (ref. 10. Chai et al., Circ Res, 2005). The present study has been conducted to understand why the vascular wall of HAS-2 mice is more susceptible to the development of atherosclerosis. Therefore, the mice are studied in the pre-atherosclerotic stage.

  1. While reading the article, it remains unclear how the authors accessed endothelial dysfunction in mice included in the experiment. If the authors draw a conclusion about the development of endothelial dysfunction based on the assessment of eNOS protein expression, this is a fundamentally wrong approach. A number of both in vivo and in vitro studies show that the role of eNOS in the development of endothelial dysfunction is significantly exaggerated (it has been shown that endothelial dysfunction caused by different triggers is associated with increased of many molecular markers of endothelial dysfunction, excepting eNOS, please refer to Shishkova D.K. et al. Calciprotein Particles Link Disturbed Mineral Homeostasis with Cardiovascular Disease by Causing Endothelial Dysfunction and Vascular Inflammation. International Journal of Molecular Sciences, 2021, 22(22):12458, DOI:10.3390/ijms222212458), so it is not correct to use this marker alone. I recommend use the panel of endothelial dysfunction markers proposed by Dr. Kutikhin et al. (Kutikhin A.G. et al. Endothelial dysfunction in the context of blood-brain barrier modeling. Journal of Evolutionary Biochemistry and Physiology, 2022, 58(3):781-806, DOI:10.1134/S0022093022030139) to assess this pathological condition and analyze these markers not only at the proteomic, but also at the genetic (gene expression profiling) level.
  • The endothelial function in the present study is measured by evaluating the endothelium-dependent vasodilatation, which we find is impaired in the aorta of HAS-2 male mice. Endothelium-dependent vasodilatation in the aorta is mainly mediated by nitric oxide (NO), and therefore, we have measured NO concentrations, which is indeed decreased in the aorta of HAS-2 male mice. Myograph is the golden standard for testing endothelium-dependent vasodilation. Please see page 2, lines 84-89, and page 5, lines 157-158.
  • We agree with the reviewer that eNOS expression by itself is not a reliable marker for endothelial cell function. Functional approaches combined with relevant biomarkers are more sensitive for evaluating endothelial/vascular function.
  1. The similar problems with the assessment of atherosclerosis – authors used TG2 and LDL receptor expression, but it is not marker of atherosclerosis. Authors must perform the Oil Red staining of mice aorta to for quantification of atherosclerosis lessions (Andrés-Manzano M.J. et al. Oil Red O and Hematoxylin and Eosin Staining for Quantification of Atherosclerosis Burden in Mouse Aorta and Aortic Root. Methods Mol Biol, 2015, 1339:85-99, DOI:10.1007/978-1-4939-2929-0_5).
  • We are familiar with the methods for the evaluation of atherosclerotic lesions in mice (Chai et al., Circ Res, 2005; Buus et al., Eur J Pharmacol, 2011). In the present study, we have not aimed at inducing atherosclerosis alone to study the pre-atherosclerotic stage of the vascular wall in the HAS-2 upregulated mice. The aim of measuring TG2 and LDL receptor expression was to clarify whether altered expression of these signal pathways may increase the susceptibility of the vascular wall to development of atherosclerosis. This is now clarified in the manuscript. Please see page 10 lines 264-267.
  1. The age of mice at 4-5 months is too young to develop signs of atherosclerosis, which could be experimentally fixed, given that the mice received a normal (not hyperlipidemic) diet. I recommend revising the design of the study, changing the diet, and raising the age of withdrawal of mice from the experiment, which will allow to get more correct results.
  • As mentioned above, we do not aim to induce atherosclerosis in the mice, but to understand the impact HAS-2 upregulation has on the vascular wall making it susceptible to development of atherosclerosis.
  1. The quality of blot images is too low (high background, out of focus, fuzzy images). Please improve it.
  • We have now provided more focused images of the blots in Figure 5 and Figure 9. Please also consult the supplementary file with the full blots.

Reviewer 2 Report

The authors analyzed the endothelium in the HAS-2 transgenic mice characterized by increased stiffness of the media. They found that hyaluronan increases oxidative 30 stress in the vascular wall and that increased endothelial cell distance is associated with sex-specific endothelial dysfunction and LDL receptor upregulation.

The results are potentially interesting but the text is quite complex and difficult to read. I would suggest a revision of the manuscript trying to better describe the data.

In addition, I have some other comments:

The images in the western blot are sometimes quite fuzzy (for example Fig 5). Please, provide better images.

Fig 8. Does the increased distance among ECs reflect a change in cell morphology?

I would suggest the author to evaluate the EC cytoskeleton structure.

Do the authors have any information on EC permeability in the vessel?

Author Response

The authors analyzed the endothelium in the HAS-2 transgenic mice characterized by increased stiffness of the media. They found that hyaluronan increases oxidative 30 stress in the vascular wall and that increased endothelial cell distance is associated with sex-specific endothelial dysfunction and LDL receptor upregulation.

The results are potentially interesting but the text is quite complex and difficult to read. I would suggest a revision of the manuscript trying to better describe the data.

  • We thank the reviewer for the positive comments and suggestions. We have carefully gone through the text and rewritten it in several places. We hope the text is now more legible.

In addition, I have some other comments:

The images in the western blot are sometimes quite fuzzy (for example Fig 5). Please, provide better images.

  • We have now provided more focused images of the blots in Figure 5 and Figure 9. Please also consult the supplementary file with the full blots.

Fig 8. Does the increased distance among ECs reflect a change in cell morphology?

I would suggest the author to evaluate the EC cytoskeleton structure.

  • No obvious changes in cell morphology in the aorta from HAS-2 transgenic versus wild-type mice exist. It is an excellent suggestion to evaluate the cytoskeleton structure of the endothelial cells. We have now mentioned this possibility in the discussion on page 13, lines 373-374.

 Do the authors have any information on EC permeability in the vessel?

  • We also find it interesting to obtain information on the EC permeability in the vessels. However, it requires an entirely different approach and protocols (infusion of marked LDL); therefore, we plan to address LDL uptake in future studies with colleagues with experience in this technique.

Round 2

Reviewer 1 Report

The authors tried to correct their manuscript and provided answers to my previous remarks, but the fundamental changes to the manuscript that were necessary for its publication have not still made.

1. In their response to my remarks, the authors wrote that they used mice in the pre-atherosclerotic stage, thus confirming that they cannot speak with certainty about the development of atherosclerosis. Atherosclerosis is a multifactorial disease with a lot of pathogenetical factors. The studying only HAS-2 expression alone does not allow to conclude about atherosclerosis development without studying its clinical manifestations (Oil Red staining as I was mentioned previously). At the same time, authors used the phrase "susceptibility of the vascular wall to atherosclerosis" through the manuscript text that is more correct in the context of the obtained results.

2. The same applies to endothelial dysfunction. I still do not agree that the authors can conclude the development of endothelial dysfunction based on the results of the assessment of only one marker (eNOS). Endothelial dysfunction is a complex pathological condition and the changing of eNOS expression is only one pathological process involved in this condition. At the same time, the authors use the phrase "endothelium-dependent vasodilatation" in the text, which is fully consistent with the obtained results and approaches.

3. I fully understand the design of this study, and I have no questions about the scientific significance of its results, but, in my opinion, the authors make too fundamental conclusions that are not experimentally confirmed, and confuse the concepts of atherosclerosis and sensitivity to the development of atherosclerosis, as well as endothelial dysfunction and endothelium-dependent vasodilatation. I can advise the authors to rewrite the text using a more appropriate description of the processes they studied.

4. The blot data is slightly improved, but the quality is still poor.

Author Response

We thank the reviewer for helpful comments and suggestions.

In this resubmission the authors have significantly improved the quality of their manuscript. However, two points need to be addressed before publication:

I would strongly advise the authors to rewrite their abstract to produce a more clear and simple text. The abstract is very extensive and goes into detailed accounts that are best suited for the article’s result and discussion sections. I suggest the author reduces this section to keep only the most important elements.

Ø  We have rewritten the abstract and focused on the main points. Please see lines 19-33.

In general, it is fairly well written but does suffer from issues of English and some spelling errors. It would benefit from English editing to make it more impactful and, more importantly, clearer.

Ø  We have carefully revised and edited the text.

Minors:

Please define and use symbols and abbreviations such as Ca2+ instead of “calcium”, h instead of hours, min instead of minutes……

Ø  We have exchanged calcium for Ca2+ and abbreviated hours and minutes.

Reviewer 2 Report

I have no further comments.

I would only revise the text that is still quite complex to read.

Author Response

We thank the reviewer for the helpful comments and suggestions.

Round 3

Reviewer 1 Report

The authors still show that they are investigating atherosclerosis and endothelial dysfunction (judging by the title and conclusions). The quality of the blots is still unsufficient. Moreover, the authors did not respond to my comments. I consider that the article is still not acceptable for publication due to the facts that were presented in my first and second reviews. Authors must either reconsider unsubstantiated conclusions or do additional experiments to confirm the conclusion they make in their article.

Author Response

  • We thank the reviewer for the comments.
  • We have reworded the title to: “Sex-dependent impairment of endothelium-dependent relaxation in aorta of mice with overexpression of hyaluronan in tunica media.”
  • We have provided other immunoblots in connection with the first revision of the manuscript.
  • The discussion of the study has been rewritten and limitations addressed in a separate paragraph. Please see lines 407-428.
  • We have reworded the conclusions of the abstract. Please see lines 32-34.
  • The study has undergone external MDPI editing.